# Inorganic Salts and Dehydrating Agents Cooperatively Promoted Ru-Catalyzed Ethylene Methoxycarbonylation Using CO$_2$ as a CO Surrogate

**Meijiao Qi, Tianli Dong, Yu Kang, Li Zhang, Zhongyu Duan and Binyuan Liu ***

Hebei Key Laboratory of Functional Polymer, School of Chemical Engineering and Technology,
Hebei University of Technology, Tianjin 300130, China; meijiaoq@163.com (M.Q.); 130720637079@163.com (T.D.);
kangyuhebut@163.com (Y.K.); zllbowen@hebut.edu.cn (L.Z.); zyduan@hebut.edu.cn (Z.D.)
* Correspondence: byliu@hebut.edu.cn; Tel.: +86-13-821-687291

**Abstract:** The use of CO$_2$ as a CO surrogate for the carbonylation of olefin has attracted considerable attention due to its abundance, readily availability, nontoxicity, and recyclability. In this work, we describe the synthesis of methyl propionate (MPA), a key intermediate for methyl methacrylate in the commercial Lucite Alpha process, by the ruthenium-catalyzed methoxycarbonylation of ethylene with CO$_2$ as a carbonyl source. An efficient approach to producing MPA has been developed by adding metal halide promoters and dehydrating agents. Control experiments suggest that the NHC-Ru-hydride may be the real active species formed in situ by the reaction of Ru$_3$(CO)$_{12}$ with ionic liquid (IL). NMR data demonstrate that inorganic salts favor the formation of active species, which is an important issue for their promotion effect. In terms of the strategy to overcome chemical equilibrium by the addition of dehydrating agents and IL participation in the formation of NHC-Ru-hydride active species, a tasked IL containing a siloxyl group was employed to Ru-catalyze the methoxycarbonylation of ethylene, which showed higher catalytic efficiency in comparison to IL without a siloxyl group.

**Keywords:** carbon dioxide; hydroesterification; inorganic salts; methyl propionate; Ru-hydride species; the siloxyl-containing ionic liquid

## 1. Introduction

Catalytic alkene alkoxycarbonylation represents a significantly straightforward process to generate one carbon elongated esters with high synthetic versatility [1,2]. An excellent example is palladium-catalyzed formation of methyl propionate (MPA) by alkoxycarbonylation of ethylene with CO as a carbonyl source. The resulting MPA is an intermediate for methyl methacrylate in the commercial Lucite Alpha process [3]. Although CO is a cheap and ideal C1 source for large-scale carbonylation chemistry, it is difficult to handle and transport due to its highly toxic and flammable properties. Hence, considerable efforts have been devoted to developing CO-free carbonylation in the past few decades [4–7]. Among all of the CO substitutes, CO$_2$ is considered as a very interesting CO surrogate due to its abundance, ready availability, nontoxicity, and recyclability [8,9]. Related to this work, Beller and co-workers [10] successfully developed a Ru-catalyzed alkoxycarbonylation of alkenes with CO$_2$ and alcohols to give the corresponding carboxylic acid esters. The application of this novel alkoxycarbonylation reaction was further investigated by the Stouten and Wang group [11], who performed the methoxycarbonylation in continuous flow under supercritical conditions. These works demonstrated that the supercritical CO$_2$ process was found to have a remarkably better environmental performance compared to the Lucite Alpha process. Further, investigation by Ji et al. [12] elucidated the role of ionic liquids (ILs) as a hydrogen donor medium to form real active species Ru-H in situ, which facilitates the reduction of CO$_2$ to CO, promoting the Ru-catalyzed carbonylation of olefins.

Recently, Xia and co-workers [13] proposed a $[Ru(CO)_3Cl_2]_2/Co_2(CO)_8$ heterobimetallic binary catalytic system for the hydroesterification reaction of olefin with $CO_2$, reducing the usage of noble metal and ILs additives. Nevertheless, the efficiency of this promising hydroesterification reaction process with $CO_2$ is still far from satisfactory, especially for the value chemicals of MPA from ethylene methoxycarbonylations using $CO_2$ as the feedstock.

The addition of inorganic salts into the metal-catalyzed carbonylation reaction, including the alkene alkoxycarbonylation, have been found to be an attractive strategy in improving reaction efficiency and selectivity [14–16]. However, to date, some published data on the order of promotion of salts are contradictory to each other. For example, Sasaki [8] reported that the order of promotion of lithium salts was $Cl^- > Br^- > I^-$ in the ruthenium complex-catalyzed hydroformylation of cyclohexene with $CO_2$. On the contrary, an opposite sequence was observed in the $Ru_3(CO)_{12}$-initiated hydroesterification of alkenes with 2-pyridylmethyl formate as CO surrogate, where lithium iodide (LiI) showed the best activation effects, and the use of chloride salt resulted in a marginal improvement [17]. Furthermore, it has been observed that the inorganic salt additives play different roles in the Ru-catalyzed carbonylation reactions [12,14–24]. For instance, Tominaga et al. [18] supposed that LiCl could facilitate the deprotonation of ruthenium hydride (Ru-H) complexes to form active species for the activation of $CO_2$ upon coordination in the hydroformylation of alkenes using carbon dioxide as a carbonyl source. At the same time, Chang and co-workers [17] argued that the effect of inorganic salts on the reaction is mainly attributed to the facile dissociation of the triratheniumcarbonyl precursor into the presumed active metal species in the Ru-catalyzed hydroesterification of alkenes and alkynes using a 2-pyridylmethyl formate as CO surrogate. From another point of view, inorganic salts additives can also promote the reduction of $CO_2$ to secondary CO, which may be realized because inorganic salts can promote the incorporation of $CO_2$ into Ru-H to form Ru-COOH, thereby activating $CO_2$ [12,14,19]. Therefore, the dependence of the type of inorganic salts on the promotion and essential role should be further clarified.

Screening the ethylene methoxycarbonylations with $CO_2$ (Scheme 1), MPA is formed together with production of $H_2O$ [10]. It is well known that esters can be hydrolyzed in the presence of $H_2O$ at high temperature, which could result in low MPA yield. As a consequence, we conceive that it is desirable to achieve a higher yield by the removal of in situ water formation and overcoming the equilibrium limitation. Such a strategy has been applied in the synthesis of dimethyl carbonate (DMC) starting from $CO_2$ and methanol [25–29]. The significance of fabricating an effective dehydrating agent for the removal of water from such a reaction is adopted for the enhancement of DMC yield. So far, various kinds of materials, including 2,2-dimethoxy propane [26], acetonitrile [27], 2-cyanopyridine (2-CP) [28], metal oxides [29], etc., have been employed as dehydrating agents to overcome this issue. Moreover, recyclable dehydrating agents are more preferable and practical in using this method to enhance yield.

$$3CH_2{=}CH_2 + 2CO_2 + 4CH_3OH \longrightarrow 3 \underset{O}{\overset{O}{\bigtriangleup}}_O{-} + 2\,H_2O$$

**Scheme 1.** Catalytic hydroesterification of ethylene with $CO_2$.

Here, we report the Ru-catalyzed methoxycarbonylation of ethylene with $CO_2$ as a CO surrogate assisted by inorganic halide additives and dehydrating agents. The roles of additives and dehydrating agents were preliminarily investigated. In light of the understanding of tetramethoxysilane (TMOS) and imidazolium ionic liquid to promote reaction, a siloxyl-functionalized ionic liquid, namely, 1-(3-trimethoxysilylpropyl)-3-methyl imidazolium chloride ([TmsPmim]Cl) was also used in this Ru-catalyzed methoxycarbonylation of ethylene with $CO_2$.

## 2. Results and Discussion

### 2.1. Promoting Effect of Dehydrating Agent and Inorganic Salts

From the viewpoint of chemical reaction equilibrium, using dehydrating agents to remove by-product $H_2O$ from the reaction mixture should be an effective strategy to promote such transformation (Scheme 1). Therefore, the influence of several dehydrating agents on the $Ru_3(CO)_{12}$-catalyzed methoxycarbonylation of ethylene using $CO_2$ as a C1 surrogate was examined with the aim of increasing the productivity of MPA. The results are summarized in Table 1. Notably, 2-CP, an efficient dehydrating agent in the synthesis of DMC directly from $CO_2$ and $CH_3OH$, deteriorated the synthesis of MPA, as evidenced by the lower yield than that without 2-CP (entry 4 vs. entry 1, Table 1). This most likely resulted from the fact that 2-CP can competitively coordinate with the Ru active center, limiting activation of the reactant by coordination interaction. To verify our hypothesis, we treated a solution of $Ru_3(CO)_{12}$ with 2-CP at the molar ratio of 2-CP: $Ru_3(CO)_{12}$ = 100:1 at 200 °C for 30 min and the solution was subjected to IR measurement. As shown in Figure S1, the original carbonyl peaks of $Ru_3(CO)_{12}$ shifted from 1960 and 2035 $cm^{-1}$ to high wavenumber (2020 and 2060 $cm^{-1}$) after $Ru_3(CO)_{12}$ treated by 2-CP. Furthermore, a shift of CN stretching vibration frequencies also occurred. These results confirmed that a coordination interaction probably takes place between $Ru_3(CO)_{12}$ and 2-CP.

**Table 1.** Effects of dehydrating agents on hydroesterification reaction [a].

| Entry | $Ru_3(CO)_{12}$ [mmol] | Ionic Liquids | Time [h] | Dehydrating Agent | Yield [%] [b] | TOF [$h^{-1}$] [c] |
|:-----:|:-----:|:-----:|:-----:|:-----:|:-----:|:-----:|
| 1 | 0.042 | [Bmim]Cl | 2 | - | 54.8 | 10.6 |
| 2 | 0.042 | [Bmim]Cl | 2 | TMOS | 73.4 | 14.2 |
| 3 | 0.042 | [Bmim]Cl | 2 | TEOS | 69.9 | 13.6 |
| 4 | 0.042 | [Bmim]Cl | 2 | 2-CP | 21.1 | 4.1 |
| 5 | 0.010 | [Bmim]Cl | 4 | - | 40.5 | 16.5 |
| 6 | 0.010 | [TmsPmim]Cl | 4 | - | 61.1 | 23.9 |
| 7 | 0.010 | [Bmim]Cl | 4 | TMOS | 72.1 | 29.1 |

[a] Reaction conditions: Ionic liquids (5.6 mmol), dehydrating agent (4.2 mmol), Methanol 5.6 mL, Ethylene 0.2 MPa, $CO_2$ 4.0 MPa, 200 °C. [b] Yield was determined by GC analysis using n-heptane (40 μL) as an internal standard. [c] Turnover frequency as moles of MPA per mole of catalyst per hour.

To our delight, the productivity of MPA was enhanced by the addition of tetraalkoxysilanes such as TMOS (tetramethoxysilane) and TEOS (tetraethyloxysilane). When TMOS was used as the dehydrating reagent, the MPA yield was 18.6% higher than without TMOS, reaching 73.4% with the presence of 0.042 mmol $Ru_3(CO)_{12}$ and 5.6 mmol [Bmim]Cl (entry 1 vs. entry 2, Table 1). A slight decrease of MPA productivity was observed when TEOS was used as the dehydrating reagent (entry 3, Table 1). The difference in efficiency between TMOS and TEOS can probably be attributed to the fact that TMOS is prone to hydrolysis [30], and methanol, which is a reactant, is released. From the point of chemical equilibrium, such a case is beneficial to the reaction toward formation of the product side. Another promising prospect with TEOS as a dehydrating agent is that TMOS can be effectively synthesized through a base-catalyzed reaction of $SiO_2$ and methanol in the presence of $CO_2$ and acetal [31]. More interestingly, $SiO_2$ can be obtained from rice hull ash, which is a natural and sustainable silica substrate [32].

The addition of inorganic salts into the metal-catalyzed carbonylation reaction, including the alkene alkoxycarbonylation, has been found to be an attractive strategy in improving reaction efficiency and selectivity [14–24]. Accordingly, four kinds of inorganic salts of LiI, CuCl, LiF, and $MgCl_2$ were selected for the preliminary evaluation of the effect of salts on the methoxycarbonylation of ethylene with $CO_2$. Interestingly, the efficiency to selectively generate MPA was substantially improved for all additive salts tested, the extent of which varied depending on the type of salts. Here, LiI provided the highest promoting efficiency, improving by about 20% (Figure 1). This result is different from the previous report regarding the ruthenium-catalyzed one-pot hydroformylation of alkenes using $CO_2$ as a reactant, where LiI showed the worst performance [18]. Considering the environmen-

tally friendly nature and satisfactory acceleration of $MgCl_2$ in the methoxycarbonylation of ethylene with $CO_2$, unless otherwise specified, the subsequent reactions were conducted using $MgCl_2$ as promoter.

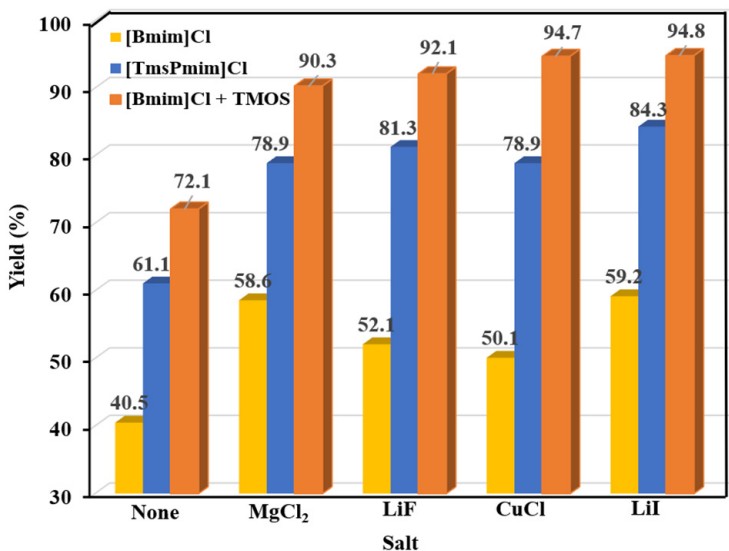

**Figure 1.** Effect of salt on the productivity of ethylene methoxycarbonylation with $CO_2$ as CO surrogate. Reaction conditions: $Ru_3(CO)_{12}$ (0.010 mmol), Salt (0.126 mmol), Methanol 5.6 mL, Ethylene 0.2 MPa, $CO_2$ 4.0 MPa, 200 °C, 4 h. (■) [Bmim]Cl (5.6 mmol); (■) [TmsPmim]Cl (5.6 mmol); (■) [Bmim]Cl (5.6 mmol) and TMOS (4.2 mmol).

### 2.2. Exploration of Reaction Mechanism

Ru-hydride-carbonyl-carbene complexes have been regarded as possible catalytic active species in the hydroformylation of alkenes using $CO_2$ as the carbonyl source [15,33,34]. To explore the possible formation of such species, the reaction mixture of $Ru_3(CO)_{12}$ and [Bmim]Cl in the absence or presence of $MgCl_2$ additive was investigated by NMR spectroscopy in light of reports (Figure 2) [15]. In the absence of $MgCl_2$ additive, it was observed by $^{13}C$ NMR spectra that the intensity of the signal at 163 ppm, assigned to N-heterocyclic carbene (NHC), enhanced gradually. Furthermore, the intensity peak at about −18 ppm, which is assignable to ruthenium hydride, showed a similar trend with prolonging reaction time. These results suggested the formation of Ru-hydride-carbonyl-carbene complex, face-capped trinuclear $[Ru_3(\mu\text{-}H)_2(\mu_3\text{-MeImCH})(CO)_9]$ (**I**) [15,33]. The isolated **I** displayed a slightly higher activity than pristine $Ru_3(CO)_{12}$ in the methoxycarbonylation of ethylene with $CO_2$ under identical conditions (entry 1 vs. entry 2, Table 2). This observation supports **I** as a possible active species generated in situ from the combination of $Ru_3(CO)_{12}$ with the [Bmim]Cl catalytic system. The easier formation of active species **I** in the presence of $MgCl_2$ is responsible for the promoting effect because metal halide salt can interact with $Cl^-$ anion of imidazolium ionic liquids. The C2-H bond can be weakened by the addition of inorganic salts and then 2-H is easy to disengage from the 1-alkyl-3-methylimidazolium-based IL [35,36]. This facilitates the formation of **I** carbene species. This speculation is further supported by the fact that the intensity of signals characteristic of Ru-NHC and Ru-H in the presence of $MgCl_2$ at 5 min are comparable to those without $MgCl_2$ for 60 min (Figure 2). The interactions varied with the kind of metal salt additive, which accounts for the observation of the dependence of yield (or activity) on the metal salt employed. It is the best evidence that the adduct of $BmimMgCl_3$ derived from $MgCl_2$ and [Bmim]Cl demonstrated a similar promoting effect to the binary $MgCl_2$ and [Bmim]Cl in the Ru-catalyzed hydroesterification reaction (entry 6 vs. entry 7, Table 2).

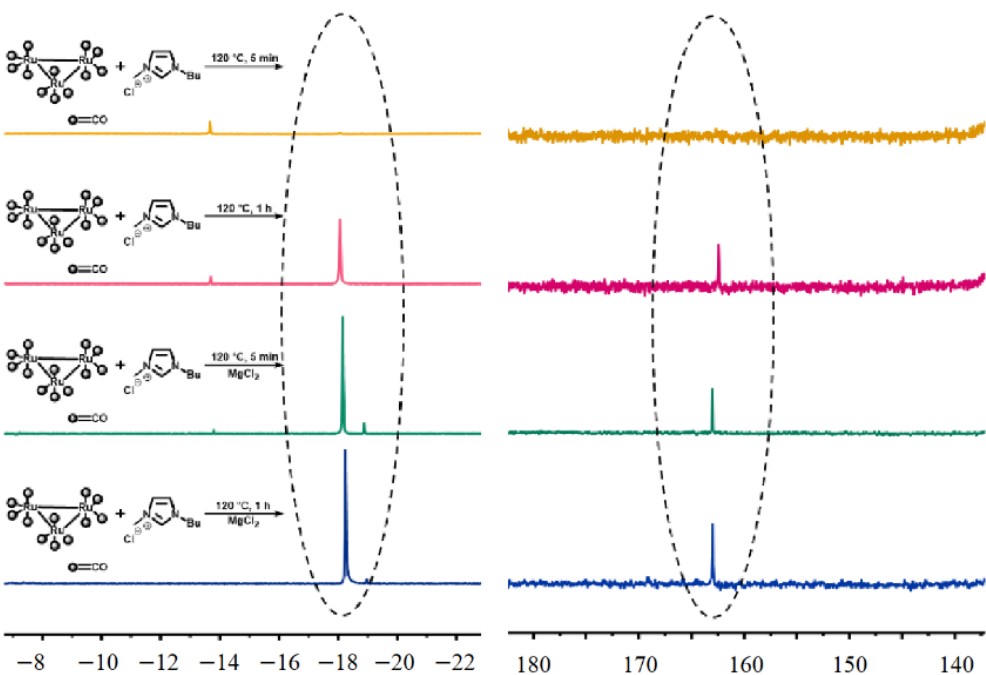

**Figure 2.** $^1H$ (400 MHz, $CDCl_3$) NMR and $^{13}C$ (100 MHz, $CDCl_3$) NMR partial spectra of the species formed by reacting $Ru_3(CO)_{12}$ with [Bmim]Cl. Reaction condition: $Ru_3(CO)_{12}$ (0.2 mmol), [Bmim]Cl (3.2 mmol), 120 °C.

**Table 2.** Hydroesterification reaction performed under various catalytic species [a].

| Entry | Carbene [mmol] | $Ru_3(CO)_{12}$ [mmol] | [Bmim]Cl [mmol] | Salt | Yield [%] [b] |
|---|---|---|---|---|---|
| 1 | 0.042 | - | - | - | 28.4 |
| 2 | - | 0.042 | - | - | 21.1 |
| 3 | 0.042 | - | - | $MgCl_2$ | 40.2 |
| 4 | - | 0.042 | - | $MgCl_2$ | 34.3 |
| 5 | - | 0.042 | 5.6 | - | 55.6 |
| 6 | - | 0.042 | 5.6 | $MgCl_2$ | 63.4 |
| 7 | - | 0.042 | 5.6 | $BmimMgCl_3$ | 62.3 |

[a] Reaction conditions: [Bmim]Cl (5.6 mmol), Salt (0.126 mmol), Methanol 5.6 mL, Ethylene 0.2 MPa, $CO_2$ 4 MPa, 200 °C, 4 h. [b] Yield was determined by GC analysis using *n*-heptane (40 μL) as an internal standard.

It should be pointed out that the promoting effect of the salt additive is also probably due to the facile reduction of $CO_2$ by Ru-H to CO, as evidenced by Han and co-workers through a computational and experimental study. [12] Further indirect proof of results is from entry 3 in Table 2, where the yield of MPA was further improved by addition of $MgCl_2$ into the Ru (**I**) catalytic system. Accordingly, a plausible mechanism is proposed as shown in Scheme 2. The whole reaction starts with the active species Ru (**I**) produced by the reaction of $Ru_3(CO)_{12}$ and [Bmim]Cl. The addition of inorganic salts can promote the formation of Ru-hydride-complexes. In this process, metal halides can promote the reduction of $CO_2$ by forming Ru-COOH intermediates [12]. The conversion rate of $CO_2$ to CO is positively correlated with the efficiency of hydroesterification. Further, in light of this assumption, the dehydrating agent plays a part by reacting with $H_2O$ produced during $CO_2$ reduction, in favor of the formation of species **III** through the insertion of CO into the Ru-R bond of species **II**.

**Scheme 2.** A probable mechanistic pathway for the Ru-catalyzed methoxycarbonylation of ethylene with $CO_2$.

### 2.3. Optimization of Reaction Conditions

To further maximize formation of MPA, the influence of catalyst loading, reaction temperature, $CO_2$ pressure, and reaction time was investigated in the presence of TMOS. As noted in Table 3, the higher temperature is favorable for the reaction. As the temperature varied from 160 °C to 200 °C, the yield increased from 27.1% to 97.5% (entries 1–3, Table 3). Furthermore, pressure plays a critical role in the catalytic activity. As indicated by entries 4 and 5 in Table 3, the yield increased significantly from 50.5% to 90.0% with increasing pressure from 2.0 MPa to 3.0 MPa. However, the productivity was promoted slightly as the $CO_2$ pressure was further raised up to 4.0 MPa. One possible explanation for the increase of yield at higher pressure is due to the increased solubility of $CO_2$ in the ILs [37,38], which leads to a higher $CO_2$ concentration in the medium, which is beneficial to the methoxycarbonylation reaction. The limited solubility of $CO_2$ in the ILs could account for the slight improvement in the reaction rate above a certain pressure (herein, 3.0 MPa). Although reducing catalyst loading from 0.042 mmol to 0.010 mmol decreases monomer conversion, the catalytic activity is still acceptable, with a yield of more than 70% in 4 h. Figure 3a shows the dependence of yields on the concentration of TMOS. The yield increases significantly with the increase of TMOS.

**Table 3.** Yield of MPA under different conditions on hydroesterification reaction [a].

| Entry | $Ru_3(CO)_{12}$ [mmol] | Temperature [°C] | Pressure [Mpa] | Yield [%] [b] | TOF [h$^{-1}$] [c] |
|---|---|---|---|---|---|
| 1 | 0.042 | 160 | 4 | 27.1 | 2.6 |
| 2 | 0.042 | 180 | 4 | 72.5 | 7.0 |
| 3 | 0.042 | 200 | 4 | 97.5 | 9.5 |
| 4 | 0.042 | 200 | 3 | 90.0 | 8.7 |
| 5 | 0.042 | 200 | 2 | 50.5 | 5.0 |
| 6 | 0.032 | 200 | 4 | 90.5 | 11.7 |
| 7 | 0.021 | 200 | 4 | 80.3 | 15.6 |
| 8 | 0.010 | 200 | 4 | 72.1 | 29.1 |

[a] Reaction conditions: [Bmim]Cl (5.6 mmol), Methanol 5.6 mL, TMOS (4.2 mmol), Ethylene 0.2 Mpa, 4 h. [b] Yield was determined by GC analysis using *n*-heptane (40 μL) as an internal standard. [c] Turnover frequency as moles of MPA per mole of catalyst per hour.

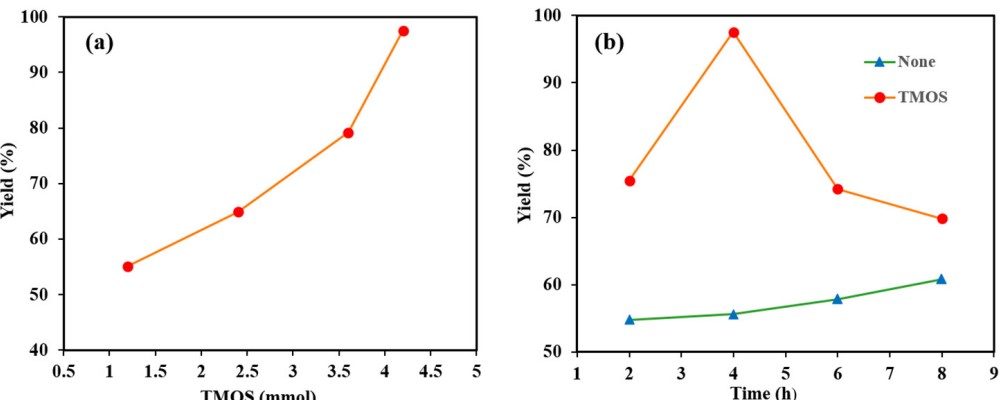

**Figure 3.** (**a**) Effect of the amount of TMOS on hydroesterification reaction of ethylene with $CO_2$. Reaction conditions: $Ru_3(CO)_{12}$ (0.042 mmol), [Bmim]Cl (5.6 mmol), Methanol 5.6 mL, Ethylene 0.2 Mpa, $CO_2$ 4.0 Mpa, 200 °C, 4 h. (**b**) Effect of reaction time on hydroesterification reaction of ethylene with $CO_2$. Reaction conditions: $Ru_3(CO)_{12}$ (0.042 mmol), [Bmim]Cl (5.6 mmol), TMOS (4.2 mmol), Methanol 5.6 mL, Ethylene 0.2 Mpa, $CO_2$ 4.0 Mpa, 200 °C.

Interestingly, with the extension of reaction time, the resulting MPA yield increased in the range of 2.0 to 4.0 h, rapidly decreased in the following two hours, and then dropped slightly. The highest yield (97.5%) was obtained when the reaction was performed at 4.0 h (Figure 3b). In the early reaction stage, the produced $H_2O$ was removed by reaction with the dehydrating agent (TMOS), and methanol and $HO-Si-(OCH_3)_3$ were formed. From the viewpoint of chemical equilibrium, this reaction is conducive to the reaction toward the generation of MPA. Meanwhile, the dimerization of $HOSi(OMe)_3$ can form siloxane $(MeO)_3Si-O-Si(OMe)_3$ and $H_2O$ by hydroxyl condensation (Scheme 3). It is well known that esters can be hydrolyzed in the presence of $H_2O$ at high temperature, which may illustrate the observed decrease in the yield of MPA at longer reaction times. This assumption was verified by the finding of a peak at $-85.7$ ppm in the $^{29}Si$ NMR spectrum for the resulting mixtures [30] (Figure S3).

**Scheme 3.** The hydrolysis and cross-linking of TMOS.

### 2.4. Applications of Siloxyl-Functionalized Ionic Liquid

To simplify the catalytic system, a siloxyl-functionalized ionic liquid, namely, 1-(3-trimethoxysilylpropyl)-3-methyl imidazolium chloride ([TmsPmim]Cl) was used to replace the TMOS and [Bmim]Cl for the Ru-catalyzed methoxycarbonylation of $CH_2=CH_2$ with $CO_2$ as carbonylation reagent. Happily, the MPA yield increased considerably from 40.5% to 61.1% under otherwise identical conditions (Table 1, entries 5 and 6). Such a phenomenon can be attributed to the fact that the siloxyl-functionalized [TmsPmim]Cl plays two roles, in participating in the formation of real active Ru species and in removing $H_2O$ via reaction with the $Si-OCH_3$ group of [TmsPmim]Cl. It should be emphasized that the catalytic system with [TmsPmim]Cl showed slightly lower activity than that with binary [Bmim]Cl and TMOS system (Table 1, entries 6 and 7). In this tasked ionic liquid-based catalytic system, the metal halides also demonstrate a promotion activity as in the ternary catalytic system comprising Ru, [Bmim], and TMOS. Again, the addition of LiI resulted in the highest activity, producing 84.3% MPA (Figure 1).

## 3. Materials and Methods

### 3.1. Materials

Unless otherwise stated, all reagents and gases were purchased from commercial suppliers and used without further purification. All manipulations involving air- or water-sensitive compounds were performed using the standard Schlenk and vacuum line techniques under argon atmosphere. $Ru_3(CO)_{12}$ (triruthenium dodecacarbonyl), [Bmim]Cl (1-Butyl-3-methylimidazolium chloride), 2-CP (2-Cyanopyridine), TMOS (Tetramethyl orthosilicate), and TEOS (Tetraethyl orthosilicate) were purchased form Heowns Biochem Technologies LLC (Tianjin, China) and used directly. All salts were purchased from Tianjin Kemiou Chemical Reagent Co Ltd. (Tianjin, China), and dried under a vacuum at 80 °C overnight before use.

### 3.2. Characterization

All $^1$H, $^{13}$C, $^{29}$Si NMR spectra were recorded on a Bruker-400 spectrometer (Bruker, ZÜRICH, Switzerland) at room temperature with reference to tetramethylsilane (TMS) as internal standard in $CDCl_3$. FT-IR spectra were measured on a Bruker Vector 22 spectrometer (Bruker, Ettlingen, Germany). The spectra of the compounds were acquired in 400–4500 cm$^{-1}$ wavenumber range using a KBr pellet. ESI-MS experiments were carried out using a Bruker Q-TOF mass spectrometer (Bruker, Ettlingen, Germany). Samples were inserted into the electrospray interface at a flow rate of 2 mL/min and the source temperature was kept at 200 °C. UV-visible spectra were collected on a G6860A spectrophotometer (Agilent Technologies, Mulgrave, Australia). Products were identified with GC analysis on a SP-6890 instrument (Lunan Ruihong, Shandong, China).

### 3.3. Synthesis of 1-(3-Trimethoxysilylpropyl)-3-Methylimidazolium Chloride ([TmsPmim]Cl)

According to previous literature [39], the synthesis route of [TmsPmim]Cl is shown in Scheme 4. The mixture of *N*-methylimidazole (8.21 g, 0.1 mol) and 3-trimethoxysilylpropyl chloride (19.87 g, 0.1 mol) was stirred and heated to 343 K for 24 h under an argon atmosphere. After cooling down to room temperature, the mixture was washed with ethyl acetate five times. The solvent was removed on a rotary evaporator and the sample was dried in vacuum for 12 h to afford a yellow viscous liquid (92% yield). The $^1$H-NMR spectrum of compound is shown in Figure S4.

**Scheme 4.** Preparation of [TmsPmim]Cl.

### 3.4. Synthesis of Metal-Based Ionic Liquids: BmimMgCl$_3$

The BmimMgCl$_3$ was prepared by mixing equal moles of [Bmim]Cl and MgCl$_2$[40]. The compound [Bmim]Cl (3.49 g, 20 mmol) was dissolved in dry acetonitrile (10 mL) and MgCl$_2$ (1.90 g, 20 mmol) was added. The mixture was stirred and refluxed for 24 h under argon atmosphere. The resulting hot solution was filtered immediately and treated by a rotary evaporator to remove acetonitrile, before being dried in a vacuum oven at 333 K overnight to yield the final product, BmimMgCl$_3$ (95% yield). $^1$H-NMR and ESI-MS spectra of this complex are shown in Figures S5 and S6, respectively.

### 3.5. Synthesis of N-Heterocyclic Carbene Complexes of Ruthenium

According to previous literature [33], a mixture of [Bmim]Cl (69.87 mg, 0.4 mmol) and KO$^t$Bu (44.9 mg, 0.4 mmol) was stirred in acetonitrile (5.0 mL) at room temperature for 20 min under argon atmosphere. Powdered $Ru_3(CO)_{12}$ (255.73 mg, 0.4 mmol) was then added, and immediate bubbling of the solution was observed. After 2 h, the solvent was filtered immediately to removed unreacted $Ru_3(CO)_{12}$ and treated by a rotary evaporator

to remove acetonitrile. The resulting crude product was washed with hexane (50 mL) to leave a brick red powder (75% yield).

### *3.6. General Procedure for the Hydroesterification Reaction*

All the hydroesterification experiments were carried out in a 20 mL autoclave equipped with a magnetic stirring bar. In a typical experiment, $Ru_3(CO)_{12}$ (6.7 mg, 0.01 mmol), co-catalyst [Bmim]Cl (0.98 g, 5.6 mmol), TMOS (0.64 g, 4.2 mmol), $MgCl_2$ (0.012 g, 0.13 mmol), and methanol (5.6 mL) were charged into the reactor in an atmosphere of argon. The autoclave was then pressurized with ethylene (0.2 MPa) and carbon dioxide (4.0 MPa). The autoclave was heated at the required temperature for the specified time. After reaction, the autoclave was cooled. The excess gas was released, and the reaction mixture was passed through a short pad of silica gel before analysis by GC.

The yield of MPA is defined as follows:

$$m_A = \frac{S_A m_B}{S_B F} \quad Yield(\%) = \frac{m_A}{n_{ethylene} \times M_{MPA}} \times 100\%$$

where $m_A$ is the weight of MPA analyzed by GC after reaction; $S_A$ is the peak area of MPA; $m_B$ is the weight of internal standard (*n*-heptane); $S_B$ is the peak area of internal standard (*n*-heptane); $F$ is the correction factor; $M_{MPA}$ is the molar mass of MPA.

### 4. Conclusions

In conclusion, we report an effective catalytic system $Ru_3(CO)_{12}$/[Bmim]Cl for the direct formation of MPA from ethylene, $CO_2$, and methanol with high yields under optimized reaction conditions. The dramatic enhancement effects of certain inorganic salts and dehydrating agents for the hydroesterification reaction have been demonstrated in this work. Ru-hydride species formed *in situ* by the reaction of $Ru_3(CO)_{12}$ with [Bmim]Cl were demonstrated to be active species. The inorganic salts can promote the formation of Ru-hydride species. Based on the above conclusions, the possible reaction mechanism is described. All these findings offered a promising strategy to use inexpensive, easily available, and less toxic $CO_2$ as a $C_1$ building block for important industrial carbonylation processes.

**Supplementary Materials:** The following supporting information can be downloaded at: https://www.mdpi.com/article/10.3390/catal12080826/s1, Figure S1: IR spectra of a mixture of $Ru_3(CO)_{12}$ and 2-CP; Figure S2: $^{29}Si$ NMR spectrum of the reaction mixture; Figure S3: $^1H$ NMR spectrum of [TmsPmim]Cl; Figure S4: $^1H$ NMR spectrum of $BmimMgCl_3$; Figure S5: ESI-MS spectra of $BmimMgCl_3$.

**Author Contributions:** B.L. conceived and designed the experimental work.; M.Q., T.D. and Y.K. performed the experiments; B.L., L.Z., Z.D. and M.Q. analyzed the data; M.Q. drafted the paper. All authors have read and agreed to the published version of the manuscript.

**Funding:** This work is supported by Hebei Natural Science Foundation (B2018202112 and B2022202015).

**Data Availability Statement:** The data presented in this study are available on request from the corresponding author.

**Conflicts of Interest:** The authors declare no conflict of interest.

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
