# Peer review of "Inorganic Salts and Dehydrating Agents Cooperatively Promoted Ru-Catalyzed Ethylene Methoxycarbonylation Using CO2 as a CO Surrogate"

_catalysts, doi:10.3390/catal12080826_

Round 1
Reviewer 1 Report
In the present paper the authors describe the synthesis of methyl proprionate by a Ru-catalyzed methoxycarbonylation of ethylene in ionic liquids with carbon dioxide as a carbonyl source. The major claim of the work appears to be the cooperative use of a dehydrating agent and an inorganic salt in order to enhance the catalytic activity of Ru3(CO)12. A mechanistic investigation suggested the formation of Ru-hydride species as potential active catalysts. A trimethoxysilylpropyl-functionalized imdazolium salt was also employed as both IL and dehydrating agent.
Although the chemical ideas of such study are not particularly innovative and can be found in the references cited by the same authors, the methodology contains a somewhat useful experimental suggestions about the potential MPA production by using a combination of a reagent that catches the water (like TMOS) a source of halide (like LiI) and an ionic liquid [Bmim]Cl.
I think the study is worthy of publication but requires revisions:
11) An extensive editing of English language and style is required
22) 1H and 13C NMR spectra related to the species formed by reacting Ru3(CO)12 with [Bmim]Cl (Figure 2) should be reported as full spectra in the supplementary material file. Some important regions of spectra are missing in the figure 2.
33) Detailed information about the MPA yields calculations should be added.
44) The TMOS/H2O molar ratio in Figure 3 should be clarified in a note or substituted with the simple TMOS (mmol).
55) Chemical structures in Figure 2 are too small, please make them more visible
Reviewer 2 Report
The article by Qi M. et al. discusses the synthesis of methyl propionate from ethylene using methanol, carbon dioxide, ruthenium catalyst, and ionic liquids. The authors consistently consider how the use of a dehydrating agent, an inorganic salt, and the nature of the ionic liquid affect the yield of the product and find the optimal conditions for the synthesis. In general, this article is very clearly and accurately written, addresses a relevant issue, and can be published in Catalysts after a minor revision.
The specific comments are as follows.
The title of the article is incorrectly stated in the Supporting Information.
Lines 15, 317: The following substitution is needed: “formed in suit by” -> “formed in situ by”
Line 64: “While” -> “At the same time,”
Line 100: “methaoxycarbonylation” -> “methoxycarbonylation”
Lines 101, 118: “the productivity of MPA.” -> “the productivity of MPA synthesis.”
Line 136: “the metal-catalyzed the carbonylation reaction” -> “the metal-catalyzed carbonylation reaction”
Line 140: “of the salt effect on” -> “of the salt addition on”
Line 148: “the following reactions” -> “the subsequent reactions”
Figure 2: The reaction conditions (temperature, time, presence of MgCl2) are better to indicate in the caption to the figure because they are not visible in the figure itself due to the small size of the caption font, and then it is useful to add the letters a, b, c, and d to the figures.
Line 178: “Ru-catalyzed the hydroesterification reaction” -> “the Ru-catalyzed hydroesterification reaction”
Line 185: “the facile the reduction of” -> “the facile reduction of”
Line 188: “plasusible” -> “plausible”
Line 210: “in media” -> “in medium”
Line 212: “of reaction rate exceeding a certain pressure” -> “of reaction rate at exceeding a certain pressure”
Line 213: “reducing catalyst loading” -> “the reducing of catalyst loading”
Line 229: “the resulted MPA increased in the range of 2.0 to 4.0 h and rapidly decreased in the following two hours, and then slightly” ->“the resulted MPA yield increased in the range of 2.0 to 4.0 h, rapidly decreased in the following two hours, and then dropped slightly more”
Line 233: “(TMOS) and methanol and” -> “(TMOS), and methanol and”
Line 254: “halide were also demonstrated promotion order” -> “halides also demonstrated a promotion activity”
Line 263: “methylimidazoliu chloride” -> “methylimidazolium chloride”.
